# Hepatitis B and Hepatitis D Viruses: A Comprehensive Update with an Immunological Focus

**DOI:** 10.3390/ijms232415973

**Published:** 2022-12-15

**Authors:** Daniel G. Sausen, Oren Shechter, William Bietsch, Zhenzhen Shi, Samantha M. Miller, Elisa S. Gallo, Harel Dahari, Ronen Borenstein

**Affiliations:** 1School of Medicine, Eastern Virginia Medical School, Norfolk, VA 23507, USA; 2The Program for Experimental and Theoretical Modeling, Division of Hepatology, Department of Medicine, Stritch School of Medicine, Loyola University Chicago, Maywood, IL 60153, USA; 3Independent Researcher, Chicago, IL 60645, USA; 4Division of Dermatology, Tel Aviv Sourasky Medical Center, Tel Aviv 64239, Israel

**Keywords:** HBV, HDV, HBsAg, HDAg, immune response, overview, life cycle

## Abstract

Hepatitis B virus (HBV) and hepatitis delta virus (HDV) are highly prevalent viruses estimated to infect approximately 300 million people and 12–72 million people worldwide, respectively. HDV requires the HBV envelope to establish a successful infection. Concurrent infection with HBV and HDV can result in more severe disease outcomes than infection with HBV alone. These viruses can cause significant hepatic disease, including cirrhosis, fulminant hepatitis, and hepatocellular carcinoma, and represent a significant cause of global mortality. Therefore, a thorough understanding of these viruses and the immune response they generate is essential to enhance disease management. This review includes an overview of the HBV and HDV viruses, including life cycle, structure, natural course of infection, and histopathology. A discussion of the interplay between HDV RNA and HBV DNA during chronic infection is also included. It then discusses characteristics of the immune response with a focus on reactions to the antigenic hepatitis B surface antigen, including small, middle, and large surface antigens. This paper also reviews characteristics of the immune response to the hepatitis D antigen (including small and large antigens), the only protein expressed by hepatitis D. Lastly, we conclude with a discussion of recent therapeutic advances pertaining to these viruses.

## 1. Introduction

Hepatitis B virus (HBV) is a highly prevalent virus worldwide. Indeed, it was estimated that 248 million individuals were hepatitis B surface antigen (HBsAg) positive as of 2010 [1]. HBsAg is a HBV protein that serves as a marker of HBV infection [2]. According to World Health Organization (WHO) data from 2019, the case number is rising, with 296 million individuals living with chronic hepatitis B infection and 1.5 million new infections each year. The WHO data further shows that almost 1 million persons die annually from complications of chronic hepatitis B infection [3]. The most common mode of HBV transmission is through exposure to infected blood or bodily fluids [4]. HBV is often spread perinatally at birth, during sexual contact, or through recreational injection drug use [4]. Spread in the healthcare setting is associated with exposure to contaminated blood or blood products [4], such as needle sticks [5] or transfusions [6]. While most infected adults clear the virus, 5–10% develop a chronic HBV infection and subsequent hepatic disease that can be quite severe. Examples of serious outcomes include acute hepatitis/fulminant hepatic failure, cirrhosis, and hepatocellular carcinoma [7]. Extrahepatic manifestations, such as polyarteritis nodosa, arthralgia, neuritis, vasculitis, and glomerulonephritis, may also complicate HBV infection [8,9,10]. Many of these adverse effects may be secondary to immune-mediated damage rather than the direct cytopathic effects of the virus itself [11,12,13,14]. HBV-specific effector CD8+ cells induce hepatocyte apoptosis in infected individuals. If these apoptotic cells are not quickly cleared by Kupffer cells, the apoptotic hepatocytes can release damage-associated molecular patterns that result in the recruitment of neutrophils and a subsequent damaging immune response [14]. However, it is worth noting that there is evidence suggesting HBV-mediated damage may play a role as well [15,16]. HBV and associated diseases represent a significant cause of worldwide mortality. In fact, it was estimated that HBV caused 39% of the approximately 820,000 deaths secondary to hepatic cancer and 29% of the 1.32 million deaths secondary to cirrhosis worldwide in 2017 [17]. 

HBV has been classified into at least 10 genotypes labeled A-J [18,19], as well as several subtypes. These classifications differ in several factors such as global distribution, gene expression levels, and clinical course [18,19,20,21]. For example, genotype A was associated with prolonged HBsAg elevation and development of chronic HBV following acute hepatitis infection in Japanese adults [22], and genotype C has been associated with hepatocellular carcinoma (HCC) [23]. The genotype also has implications for response to treatment. For example, genotypes A and B were shown to be more likely to seroconvert than genotype D following treatment with pegylated interferon (PEG-IFN) in hepatitis E antigen (HBeAg)-positive patients [24]. Following cessation of tenofovir disoproxil fumarate or entecavir in HBeAg- negative patients, those with genotype B are more likely to relapse and require retreatment than genotype C [25]. 

Hepatitis D virus (HDV) is considered a satellite virus because it requires HBV surface proteins to generate mature virion particles [26]. As a result, HDV can establish either co-infection or superinfection with HBV, but cannot package infectious virions or spread in the absence of HBV [26]. It was previously demonstrated that enveloped viruses distinct from HBV can form infectious delta virus particles [27], but there is currently no firm evidence for the clinical significance of these observations gleaned from cell cultures and mouse models. Coinfected individuals may develop an acute hepatitis but are typically able to clear both viruses. HDV superinfection of a chronically infected HBV carrier is more likely to result in chronic HDV infection with its concomitant comorbidities. Development of a chronic HDV superinfection can exacerbate hepatic injury caused by HBV [26,28], and both coinfection and superinfection have been shown to result in more severe outcomes than HBV infection alone, including fulminant hepatitis, HCC, and chronic hepatitis [26,29].

While not quite as prevalent as HBV, a study published in September 2020 estimated that 12 million individuals worldwide are seropositive for anti-HDV antibodies [30]. Notably, there has been some variance regarding estimates of HDV prevalence, with a separate study placing the number of infected individuals as high as 62–72 million [31], while another study published in May 2020 estimated that there are between 48–60 million HDV infections worldwide [29]. Modes of HDV transmission include exposure to contaminated blood, high risk sexual behavior, and injection drug use. [32]. The risk of vertical HDV transmission from mother to child is relatively low [33]. Chronic hepatitis D is considered the most severe of all hepatitis infections [34] and is associated with increased risks of hepatocellular carcinoma, cirrhosis, liver decompensation, and mortality [35]. Like HBV, HDV-associated disease is considered to be largely immune-mediated in nature [36].

### HBV and HDV Cell Tropism

While HBV primarily infects hepatocytes in vivo, it has been shown to infect non-hepatic cells in vivo as well, including but not limited to endothelial cells, hematopoietic precursors, neuronal cells, and monocytes/macrophages [37]. The primary cell lines used for in vitro experiments are hepatic cell lines including primary human hepatocytes, primary tupaia hepatocytes, and the hepatic cancer lines HepaRG, Huh7, and HepG2. These cell lines also support in vitro experiments involving HDV (reviewed in [38]). The discovery of sodium taurocholate cotransporting polypeptide (NTCP), a multiple transmembrane transporter expressed in the human liver, as a key entry receptor for both HBV and HDV [39] has resulted in engineered cell lines. These cell lines, such as HepG2-NTCP and Huh7-NTCP, are susceptible to infection by both viruses [38,40]. Only some primates and tree shrews are susceptible to human HBV infection, limiting in vivo experimentation. However, models of HBV infection based on related viruses exist in woodchucks, ground squirrels, and ducks [41]. Organisms naturally susceptible to HDV infection are similarly few in number and include chimpanzees and tree shrews [42]. Human liver chimeric mice supporting both HBV and HDV infection have also been developed [42,43], and mice models carrying HBV and HDV infection have been created through hydrodynamic delivery [44,45,46]. Numerous additional approaches have been taken to model HBV and HDV infection in animal models, which have reviewed recently [42,47].

## 2. Virion Structure and Viral Life Cycle of HBV and HDV

HBV is a member of the Hepadnaviridae family and belongs to the genus Orthohepadnavirus. The virion is approximately 42 nm in diameter. A host-derived membrane that contains small (S-HBsAg), middle (M-HBsAg), and large (L-HBsAg) hepatitis B surface antigens [48,49] surrounds an icosahedral capsid comprised of the core antigen [50]. HBV’s partially double-stranded, relaxed circular DNA (rcDNA) genome, which is approximately 3.2 kb in length, includes a lesion bearing a plus and minus strand. The HBV polymerase is covalently attached via a 10 nucleotide DNA flap to the minus strand. The plus strand contains a gap and an RNA primer which is a remnant of the reverse transcription reaction [51,52]. Figure 1a below demonstrates the structure of HBV and HDV, while Figure 1b illustrates their genomic structures.

The HBV entry process into hepatocytes can be subdivided into several steps. In the first step, the virus attaches to the surface of the host cell by binding to heparan sulfate proteoglycans (HSPGs) such as glypican 5 [55]. Following this initial attachment, HBV is thought to bind to its dedicated receptor–NTCP [39]. Notably, Perez-Vargas et al. recently showed that neither HSPGs nor NTCP were required to induce viral fusion, but that there was a key fusion peptide in the preS1 domain of L-HBsAg [56]. The authors propose that regulation of cellular cross strand disulfide bonds by the cellular protein disulfide isomerase ERp57 positions the preS1 fusion peptide to facilitate fusion between the viral and endosomal membrane [56]. HBV is believed to enter the cell via the clathrin-mediated endocytosis pathway. The viral capsid is subsequently directed to the nucleus [57], where its rcDNA is modified by cellular factors including proliferating cell nuclear antigen, replication factor C complex, flap endonuclease 1, DNA polymerase δ, and DNA ligase 1 [58] and is eventually transformed into cccDNA. cccDNA is then used as a template for transcription [57]. It is believed that this cccDNA is directly responsible for establishing chronicity [59]. 

Four different lengths of RNAs are then transcribed, pre-C/C, pre-S, S, and X mRNA. These transcripts are 3.5 kb, 2.4 kb, 2.1 kb, and 0.7 kb in length, respectively [51]. Pregenomic RNA (PgRNA), which is the RNA sequence produced following transcription from cccDNA [60], is then transported to the cytoplasm. Once there, the pgRNA is encapsidated with HBV polymerase, and rcDNA is created via reverse transcription [61]. Virions containing rcDNA are then enveloped, and the now infectious virions are secreted in multivesicular bodies (MVBs) [61,62]. 

HBV encodes the viral antigens HBsAg from pre-S and S mRNA and HBx, which is involved in infection and replication, from mRNA to the X gene [7,51]. HBsAg is the viral envelope protein and is encoded by the S gene [7,51]. It can be expressed as S-HBsAg, M-HBsAg, or L-HBsAg [50]. HBV also encodes HBeAg and core antigen (HBcAg) [7,51]. The C gene codes for HBcAg and a precore protein. The precore protein is then processed by the golgi and endoplasmic reticulum and secreted as HBeAg [63].

HDV is the only member of the *Deltavirus* genus and measures approximately 36 nm in diameter [64]. The core of the HDV virion is a single-stranded RNA molecule complexed with both the small and large forms of hepatitis D antigen (HDAg), the only protein encoded by HDV [64]. This core is surrounded by an envelope comprised of the HBV envelope proteins S-HBsAg, M-HBsAg, and L-HBsAg. While M-HBsAg is thought to be nonessential for HDV infectivity, both L-HBsAg and S-HBsAg are required [54]. 

Like HBV, HDV attaches to the host’s surface HSPGs [65] and uses NTCP for cell entry [39]. The ribonucleoprotein core travels to the hepatocyte nucleus for replication. There, the viral RNA genome replicates, generating approximately 300,000 new viral copies as well as a smaller number of complementary RNA antigenome [64], which is a replication intermediate rather than mRNA [66]. This is edited during the replication step by the cellular enzyme adenosine deaminase acting on RNA (ADAR) [67,68] to produce a second distinct antigenome strand [54] that contains the open reading frame (ORF) for HDAg [69]. Replication is accomplished in a rolling-circle fashion, and autocatalytic ribozymes present in both the genome and antigenome function to cleave the strands into linear RNA strands that are ligated into the circular antigenome RNA [70]. As HDV does not have its own RNA polymerases, replication is presumably accomplished by hijacking host enzymes [64,71]. There is supporting evidence that RNA polymerase II is used to transcribe HDV mRNA [72,73]. Both RNA polymerase I and III are known to bind to HDV RNA [74]. While previous experiments have linked RNA polymerase I with the antigenome, the function of RNA polymerase III remains undiscovered [71,74,75]. Figure 2 illustrates the viral cycles of HBV and HDV. 

HDAg is ultimately translated into two isoforms: small HDAg (S-HDAg) and large HDAg (L-HDAg). This process begins during replication, with the unedited antigenome strand producing mRNA that is translated into S-HDAg and another strand that is edited by ADAR1 to produce L-HDAg mRNA [54,76,77]. Specifically, ADAR1 alters a UAG (stop) codon to UGG (tryptophan), which allows for the expression of the larger L-HDAg [54,76,78]. S-HDAg functions to upregulate RNA production and is thought to recruit the host RNA polymerase for replication. This is because it binds to RNA polymerase II [64], possibly by way of chromatin remodeling complex recruitment. It has also been shown to interact with polymerase I. S-HDAg is required for replication [54]. In contrast, L-HDAg undergoes the post-transcriptional modification prenylation [54,79], which inhibits further RNA accumulation and contributes to the assembly of new viral particles [54].

Ribonucleoprotein assembly occurs in the hepatocyte nucleus. Assuming the viral particle can obtain the necessary HBV envelope proteins, the infectious delta virions are thought to exit the cell through the golgi [27,54].

## 3. Natural Course of HBV and HDV Infection

According to guidelines established by the American Association for the Study of Liver Diseases, there are four phases of chronic hepatitis B infection [80], All four phases are defined by the presence of HBsAg for ≥6 months. In the first phase of chronic HBV infection, serum HBV levels range from undetectable to several billion IU/mL. Patients in this phase are subdivided into either HBeAg positive or negative. HBV-DNA levels are often > 20,000 IU/mL in HBeAg-positive chronic HBV, while lower values of 2000–20,000 IU/mL are typically seen in HBeAg-negative chronic HBV patients. ALT and AST levels can be either normal or elevated, and liver biopsy demonstrates chronic hepatitis with variable necroinflammation and/or fibrosis [80]. Patients in the second phase, known as the Immune-Tolerant (IT) phase, are HBeAg positive with elevated HBV DNA levels often >1 million IU/mL. ALT/AST levels are normal or slightly elevated. Liver biopsy performed during this stage demonstrates minimal inflammation with no fibrosis [80]. Patients in the third phase, known as Immune-Active (IA) chronic HBV, have differing HBV DNA levels based on their HBeAg status. HBeAg-positive patients have serum HBV DNA levels > 20,000 IU/mL, while HBeAg-negative patients have lower levels of at least 20,000 IU/mL. ALT/AST levels at this stage are sporadically or persistently elevated consistent with host immune system-mediated hepatic injury [80]. Liver biopsy or noninvasive testing at this stage reveal chronic hepatitis and moderate to severe necroinflammation. Fibrosis may or may not be present [80]. The fourth and final phase is termed inactive chronic HBV. Patients in this phase are HBeAg negative and anti-HBe positive. Serum HBV DNA levels are typically <2000 IU/mL, and ALT/AST levels are firmly within normal ranges. Liver biopsy at this stage demonstrates varying degrees of fibrosis without significant necroinflammation [80]. An important consideration is that chronic HBV is dynamic; patients may transition between phases in either direction [81]. 

The risk of developing a chronic infection is not uniform; instead, it is inversely proportional to age [82,83] For example, the risk of HBV transmission from mother to baby is quite high [84]. For vertically infected infants whose mother is positive for both HBsAg and HBeAg, the risk of progression to chronic HBV infection is between 85–90% in the absence of passive-active prophylaxis with hepatitis B immune globulin and the HBV vaccine [85]. These rates decrease for children younger than 5 to between 20–50% and are the lowest for older children and adults, with between 5–10% progressing to chronic HBV [82]. 

There are two distinct patterns of infection for HDV: co-infection, which involves simultaneous infection with HBV and HDV, and superinfection, which involves the infection of an HBsAg-positive patient with HDV [86]. The pattern of infections typically influences the clinical outcome. Coinfection typically causes a more severe hepatitis with a greater incidence of acute liver failure than acute HBV infection alone. It also may lead to biphasic peaks in AST/ALT. [87]. However, patients are typically able to clear coinfections without developing chronic HBV/HDV infection. On the other hand, superinfection typically results in severe acute hepatitis with subsequent progression to chronic hepatitis in over 90% of cases [88]. Because of the differences in prognosis and treatment, discriminating between acute co-infection and superinfection in these patients is critical [89].

While the pathogenesis of HDV-mediated liver damage is not well understood, the pathogenesis of HDV-related liver disease can be attributed to several factors including HDV genotype, HBV genotype, and the host immune response [90]. The clinical characteristics of HDV infection are not easily distinguished from other causes of hepatitis and must be differentiated using diagnostic tests [91]. HDV is diagnosed by screening for HDAg-specific antibodies, namely anti-HDV IgM and IgG, in HBsAg-positive individuals. Patients with the anti-HDV reagent should be screened for serum HDV RNA to differentiate between active infection (HDV RNA positive) and a decreasing serological scar (HDV RNA negative) [89].

A study conducted by Genesca et al. assessed the effects of HDV co- and superinfection on HBV replication [92]. Levels of HBV DNA were similar in patients with acute HBV infection, HBV/HDV coinfection, and HBV DNA-positive patients with HBV/HDV superinfection [92]. Of the 37 patients with HBV/HDV coinfection, one died due to fulminant hepatitis and two developed chronic infections. The remaining 34 cleared the infection within three months. One of the chronic patients developed a serological pattern featuring active infection markers of both HBV and HDV. The other patient became HBeAg and HBV DNA-negative within 4 weeks, although HBsAg and HDV antibodies were detectable for longer than two years [92]. In the superinfection group, only 25% of HBsAg carriers (6/24) had detectable HBV DNA while acutely superinfected. Chronic liver disease was documented in all 14 long-term follow up patients with superinfection. Intriguingly, HBV DNA could not be detected in 10 of these patients, although there was evidence for active HDV infection. The data indicated at least a transient inhibition of HBV replication in the presence of HDV superinfection in the four patients with initially detectable HBV DNA. In two of these patients, the time at which HBV DNA became detectable correlated with the time at which serologic markers for HDV replication became undetectable [92]. The authors concluded that patients with inactive HBV infection and HDV superinfection are more likely to have ongoing HDV replication than those with an active HBV infection at the time of superinfection [92]. 

### 3.1. HDV RNA and HBV DNA Interplay during Chronic Infection

There is limited information on the natural history of the viral kinetics of patients with chronic HBV and HDV infections. Giersch et al. [93] reported that while HDV often appears to be dominant in co-infected patients, the interplay between HDV RNA and HBV DNA can be “highly dynamic” over the course of infection. Previously, Schaper et al. [94] and later Braga et al. [95] suggested three kinetic profiles (patterns): HDV dominant, HBV dominant, and HDV/HBV equivalent. During anti-HDV treatment with lonafarnib, Mhlanga et al. [96] demonstrate a rise in HBV DNA as HDV RNA is suppressed by treatment, resulting in HBV dominance further confirming the range of kinetic profiles possible. What follows is a description of the kinetic profiles of the viruses during chronic infection within the individual and aggregate data we reviewed (Table 1). Further studies are needed for understanding the natural interplay of HBV and HDV.

### 3.2. Histopathology

Most forms of viral hepatitis, including HBV and HDV, cause necrosis and inflammation of hepatocytes in a similar and indistinguishable manner from one another [107]. In acute disease, there is widespread focalnecrosis, swelling and apoptosis of hepatocytes, and portal tract/intrasinusoidal inflammation. The inflammatory infiltrate is primarily composed of lymphocytes, although plasma cells, neutrophils, and eosinophils may also be seen [107]. Acute hepatitis pathology includes the presence of several histological features. Numerous apoptotic bodies (also called acidophil bodies or Councilman bodies) are seen, as is activation of Kupffer cells (KC, the tissue-resident macrophages of the liver [108]) in response to the viral infection [107]. In HBV, it has been shown that this activation stimulates KC to release chemokines and cytokines that attract other leukocytes [109]. Another feature of acute hepatitis pathology is ballooning degeneration, which is swelling of hepatocytes [107,110].

Distinct characteristics are likewise commonly seen in chronic hepatitis. Piecemeal/interface necrosis involves necrosis of periportal hepatocytes with accompanying inflammation involving the portal tract and periportal zone [111]. Necrosis can extend over multiple lobules, which is termed confluent necrosis. Conversely, focal necrosis involves a relatively small area of hepatocytes, with focal necrosis involving a smaller group of hepatocytes [111]. These histologic features, along with other findings common to chronic hepatitis including portal inflammation and fibrosis/cirrhosis, have been quantified [112]. The modified Histology Activity Index (HAI) [112] and METAVIR [113], which has been adapted for other forms of hepatitis besides hepatitis C virus [114], scoring systems are among the most widely used [115]. Of note, chronic HDV patients often suffer more severe histologic changes than those seen in chronic HBV patients with similar characteristics [116]. This is due to the fact that patients with chronic HDV are much more likely than chronic HBV patients to progress to more severe liver disease [65].

## 4. HBsAg, CD4+, and CD8+ T Cell-Mediated Immunity

Many types of chronic infections stimulate changes in the immune system [117,118,119]. Recent research has begun to examine the impact HBV has on the immune system, including how it impacts T cell-mediated immunity. Indeed, the importance of T cells in HBV pathogenesis is emphasized by the fact that HBV-mediated toxicity has been described as a process driven by incompletely anergized, HBV-specific CD8+ T cells that “[attack] liver cells in the attempt to eliminate HBV from the liver” [14]. Le Bert et al. characterized the effects of HBsAg on the host T cell population of those chronically infected with HBV [120]. Cytometry by time of flight and subsequent analysis on cells obtained from 48 patients with a wide range of quantitative HBsAg levels revealed that HBsAg levels did not impact the frequencies of numerous immune cells, including CD4+ T cells, CD8+ T cells, mucosal-associated invariant T cells, or NK cells. B cell and mononuclear phagocyte frequencies were also analyzed and found to be unaffected [120]. Furthermore, no change was noted in cytokine production, receptor expression, or regulatory T cell/immunoregulatory receptor levels. Interestingly, the total number of T cells specific for the entire HBV proteome was also unaffected [120]. Additional recent data showed that patients with very high (>10,000 IU/mL) levels of HBsAg had decreased frequencies of central and effector memory CD8+ T cells. Naïve CD8+ T cell levels were also found to correlate with levels of HBsAg. The authors attributed these correlations to patient age. In accordance with the above set of experiments by Le Bert et al., HBsAg levels did not appear to affect T cell function [121]. Increased levels of HBsAg did correlate with increased numbers of T cells specific for the viral envelope [120].

Age, too, has been linked to changes in the immune system [122], and studies have examined the role of aging on T cell phenotypes in the setting of HBV infection [120,121]. Age was inversely correlated with the number of T cells specific for the viral envelope, with strong responses to the HBV envelope all but limited to those under 35 [120]. Indeed, the percent of T cells specific for the HBV envelope declines from 30–40% in those aged 3–24 to less than 10% in those older than 35 [120]. The authors also analyzed 11 patients who were able to clear a chronic HBV infection. Intriguingly, these patients did not have any IFN γ-secreting T cells specific for the HBV viral envelope. In contrast, 33% of T cells from patients who cleared an acute infection were HBs-specific [120]. Likewise, Aliabadi et al. demonstrated that age has a significant effect on the T cell response to HBV. Patients younger than 40 had a higher frequency of IFN-γ+ CD4+ T cells, both overall and specific for HBV. This trend was most strongly noted in T cells specific for HBV polymerase [121]. Consistent with these results, an analysis of Chinese national serosurvey data demonstrated that HBsAg seroclearance was particularly likely in early childhood and between the ages of 20–24 and 35–39 [123]. Importantly, both Le Bert and Aliabadi concluded that age, which may act as a proxy for length of exposure to infection and thus for exposure to HBsAg, plays a significant role in altering the host immune response [120,121].

There were also differences in the T cell compartment between patients who became HBsAg negative compared to those who did not [124]. Those who were unable to clear HBV infections, as determined by persistent HBsAg levels, had lower T cell frequencies, absolute CD8+ T cells, and monocytes than healthy controls [124]. The CD4+ T cells of patients who cleared the infection had higher CD25 and Human leukocyte antigen (HLA)-DR expression. The CD8+ T cells of those who cleared the infection also had higher HLA-DR levels than both the control and those who did not clear the infection [124]. There was no change in the frequency of T cells specific for HBsAg, although there was an increase in HBcAg-specific, IFN-γ-producing CD4+ and CD8+ T cells [124]. Differences were also noted when examining T cells of those who did not clear the HBV infection but did have a decrease of at least 30% in HBsAg within 6 months v. those whose HBsAg levels remained level or increased [124]. Those who had a reduction in HBsAg had increased expression of HLA-DR and CD107a on both CD4+ and CD8+ T cells [124]. TIM-3 receptor, CD40L protein, and CTLA-4 surface molecule expression was increased on CD4+ cells and CD69 costimulatory molecule was increased on CD8+ T cells when compared to patients whose HBsAg levels did not decline [124]. 

T cell exhaustion, i.e., a state of T cell dysfunction, is associated with chronic infection and is characterized by T cells with limited effector function and chronic inhibitory receptor expression [125]. In severe cases, there may be associated deletion of virus-specific T cells [125,126]. CD4+ T cells isolated from patients with chronic hepatitis B were analyzed for the expression of inhibitory receptors consistent with T cell exhaustion. These cells were shown to have increased expression of PD-1 and LAG-3. Both PD-1+ and LAG-3+ cells produced lower levels of IFN-γ, IL-2 and TNF-α than T cells not expressing these receptors, which is consistent with T cell dysfunction [127]. Furthermore, antibodies against PD-L1 and LAG-3 stimulated TH1 cytokine production and inhibited FOXP3 [127], an essential gene in the development of regulatory T cells [128]. This is consistent with previous studies showing that PD-1 is more commonly expressed on CD4+ T cells than other inhibitory molecules including CTLA-4, TIM-3, KLRG1, and CD244 in chronic HBV infection [129]. Notably, viral clearance led to a significant decline in PD-1 expression [129]. 

CD8+ T cells can likewise develop an exhausted phenotype in the setting of HBV infection [130,131]. Thymocyte Selection-Associated High Mobility Group Box (TOX) is a transcriptional regulator whose expression is positively correlated with HBV-specific T cell stimulation by HBV antigens [132]. TOX expression did not return to baseline after chronic exposure to HBV antigens ceased. Furthermore, TOX expression by a T cell correlates with the level cell exhaustion in chronic HBV infection. TOX-expressing HBV-specific T cells had higher levels of multiple inhibitory receptors, including PD-1, KLRG1, and CD57, as well as higher expression of transcription factors associated with T cell exhaustion, including EOMES and Helios. Higher TOX levels also correlate with decreased functionality in this setting. Measures inversely correlated with TOX expression include expansion capacity, HBV-specific cytokine production, and degranulation [132]. Notably, previous studies have demonstrated that PD-L1 blockade resulted in the greatest functional improvement in exhausted CD8+ T cells [133]. PD-1 expression was highest on T cells with an intermediate level of differentiation as defined by a CD27+ CCR7- CD45RA- phenotype. These intermediately differentiated T cells demonstrated greater functional impairment than T cells at other stages of differentiation and responded more strongly to PD-L1 blockade. These results indicate that response to PD-L1 blockade was associated with an intermediate level of T cell differentiation [133].

CD4+ T cells differentiate into a variety of effector cells, including T helper (Th) 1, Th2, regulatory T cells, and follicular helper T cells. Their effector function is mediated through the secretion of cytokines [134]. CD4+ T cells play a variety of roles in infection, including memory [135] and CD8+ T cell activation [136]. In addition to their activating roles, CD4+ T cells have been linked to immune inhibition [137]. CD4+ T cells represent an avenue for immunomodulation in other chronic viruses, such as betaherpesviruses [138] and HIV [139]. Thus, it is unsurprising that HBV infection affects the CD4+ T cell compartment. Indeed, central memory CD4+ T cell levels were significantly elevated in HBV patients compared to healthy controls [140]. Additionally, expression of the inhibitory molecule PD-1 on CD4+ T cells was increased in patients with high levels (>50,000) of serum HBsAg when compared to those with low levels (<500) [140]. Likewise, CD4+ T cells with high levels of PD-1, which are associated with more significant T cell exhaustion, were increased in patients with high HBsAg levels when compared to those with low levels. Higher PD-1 levels were also noted on terminally differentiated effector (EMRA) T cells in patients with high HBsAg levels. Other inhibitory receptors whose expression was markedly elevated in patients with high HBsAg include co-expression of 2B4 and PD-1 as well as FCRL5 [140]. Consistent with these results, the frequency of CD4+ T cells secreting multiple types of cytokines was significantly higher in patients with low serum HBsAg [140]. Checkpoint blockade with αPD-L1 antibody improved CD4+ T cell responses in patients with lower serum HBsAg levels. B cell responses were also improved in those with HBsAg < 100 when compared to those with HBsAg > 5000 [140]. 

Extracellular vesicles (EVs) are secreted vesicles with variable contents such as RNA, proteins, and lipids [141,142]. Recently, EVs were shown to play a role in the CD4+ T cell response to HBsAg in vaccinated mice [143]. HBsAg-vaccinated mice were injected with EVs derived from unvaccinated mice, mice vaccinated with Ovalbumin, and mice vaccinated with HBsAg. Those mice who received EVs from HBsAg-vaccinated mice produced significantly more HBsAb. Further analysis showed an increase in the number and activity of TH1, but not TH2, cells [143]. 

### HBsAg and Follicular Helper T Cells

T follicular helper cells (Tfh) are a T cell subset that play key roles in generating and maintaining germinal centers (GC), which are found in secondary lymphoid organs. In the GC, B cells with high affinity for an antigen are expanded and differentiate into memory and plasma B cells [144,145]. Tfh cells also have a profound and direct impact on the B cells themselves, influencing proliferation, gene expression, and class switching [145]. Recent research has explored the effect of HBV on Tfh cells and its subsequent impact on the humoral immune response [146]. One such work examining the interplay between B cells, Tfh, and HBV explored differences in antibody secretion between C57BL/6N (B6N) and C57BL/6J (B6J) mice hydrodynamically transfected with the HBV plasmid pAAV/HBV1.2 [146]. B6N and B6J mice are related mouse strains with relatively minimal genetic differences (reviewed in [147]). However, experiments have demonstrated metabolic differences between the two strains, such as neonatal response to hyperoxia [148], insulin response [149], and response to chronic stress hormone exposure [150]. Mice with HBV infection established by hydrodynamic injection, such as the C57BL/6 strain [151], are commonly used models of HBV infection [44]. The antibody response and results of infection in B6N and B6J mice strains were found to be similar to acute and chronic human HBV infection, although they differed from each other [146]. While serum HBsAg and HBV DNA both were undetectable within eight weeks of transfection in B6N mice, HBsAg and HBV DNA was present in nearly 40% of B6J mice 26 weeks after transfection. Subsequent analysis demonstrated that there were fewer antibody-secreting cells specific for HBsAg in B6J mice. This difference was attributed to the absence of germinal center B cell variation secondary to a lack of Tfh expansion in B6J mice [146]. The importance of Tfh cells in generating a response to the surface antigen was demonstrated by the extremely limited hepatitis B surface antibody (anti-HBs) production seen in Tfh knockout mice. In addition to the aforementioned changes in Tfh activity, HBsAg-specific regulatory T cell (Treg) numbers were higher in HBV transfected B6J mice, and transfection of Tregs specific for HBsAg inhibited Tfh in B6N mice [146].

The frequency of Tfh cells in the setting of chronic HBV infection can vary greatly; one study encompassing 127 patients found that Tfh distribution (as defined by expression of the C-XC chemokine receptor type 5 [CXCR5] and PD-1) in CD4+ cells ranged from 12.5% to 91.3% with a median of 40%. The study noted that a significant overlap existed between cells with a Tfh and a Th1 phenotype (Th1 phenotype defined by CXCR3 expression without C-C chemokine receptor type 6 [CCR6] expression) [152]. Moreover, stimulation through OX40 and blockade of PD-L1 was found to enhance cytokine-producing CD4 T cells. There was increased production of both IL-21 and IFN- γ [152]. A separate set of experiments conducted by Khanam et al. on patient-derived cells demonstrated that circulating levels of both Tfh and several Tfh markers including PD-1 and ICOS were elevated in the context of chronic HBV infection. However, the number of HBsAg-specific Tfh cells secreting Interleukin (IL) 21 was decreased compared to vaccinated controls [153]. IL-21 is a cytokine produced by Tfh cells that broadly impacts B cell differentiation and response as well as GC functionality [154]. IL-27, a cytokine that facilitates CD4+ T cell differentiation into Tfh cells and promotes GC B cell function [155], was unaffected [153]. Surprisingly, HBsAg-specific, IL-27-secreting Tfh cells were present. These cells were primarily PD-1+ [153]. The authors showed that IL-27 secretion plays a role in stimulating B cells in the setting of chronic HBV infection [153]. ELISpot assays revealed that IL-27 augmented both HBsAg-specific and total IgG and IgM production in B cells derived from chronically infected individuals, although IgA was unaffected [153]. A similar set of experiments using B cells from vaccinated controls showed increases in IgG and total antibody production, but not IgM or IgA [153]. Chronic HBV infection was also associated with alterations in the B cell pool, including an increased percentage of CD19+ B cells as well as increases in plasmablasts, plasma cells, and regulatory B cells [153]. 

The importance of Tfh cells in mounting a response to HBsAg was highlighted by a set of experiments performed by Ayithan et al. [156]. They obtained peripheral blood samples from chronic HBV patients 8 h after receiving one dose of Selgantolimod, a Toll-like receptor (TLR) 8 agonist, that demonstrated an improved Tfh cell response. Some patients treated with selgantolimod also showed an improved HBsAg-specific IgG response. In addition, Tfh cells exposed to TLR8 agonists stimulated the production of memory and plasma B cells as well as IgG production [156]. 

## 5. HBsAg and B Cell Immunity

The potential to take advantage of the humoral immune response to hepatitis B as a therapeutic target has gained traction in recent years [157]. Indeed, humans produce anti-HBs, an antibody created in response to HBsAg following either infection or vaccination. Anti-HBs indicates infection clearance following infection and immunity following both infection and vaccination [158]. In addition, anti-HBs has long been known to confer protection against HBV infection [159,160]. Thus, a thorough understanding of the interaction between HBsAg and humoral immunity is essential. 

It was recently shown that the ability of B cells specific for HBsAg to secrete antibodies is impaired [161]. When examined in vitro, HBsAg-specific B cells were unable to expand or mature, even in the presence of CpG, a stimulus of the innate immune system. Interestingly, this dysfunction was not seen in HBcAg-specific B cells [161]. A comparative genetic analysis of B cells specific for HBcAg and HBsAg showed differential expression in numerous genes. HBsAg-specific B cells had decreased expression of genes involved in class switching, but increased expression of genes involved in endothelial adhesion, homeostasis, and apoptosis [161].

HBsAg-specific B cell dysfunction has been documented in other experiments as well [162]. Fluorescently labeled HBsAg bait was used to stain for HBsAg-specific B cells in patients. Bait-staining B cells were readily detected in chronically infected patients; however, FACS-sorted HBsAg-specific B cells from these patients were unable to produce detectable levels of antibodies against HBsAg. This contrasts sharply with B cells isolated from a vaccinated control population. Further analysis demonstrated that atypical memory B cells (AtM) were increased in chronically infected patients [162]. AtM are commonly expanded in the setting of chronic immune activation. They are characterized by morphological changes such as diminished expression of CD21 and CD27 and increased expression of numerous inhibitory receptors as well as by limitations in their response to B cell receptor stimulation [163]. This increase in AtM in chronically infected patients was more prominent in the HBsAg-specific B cell population than the global B cell population [162]. AtM receptors also promoted migration to inflamed tissue rather than lymphoid organs, precluding their ability to interact with T cells [162]. Other changes included increased AtM expression of the inhibitory receptors FcRL5, Fc_y_RIIB, PD-1, B and T lymphocyte attenuator (BTLA), and CD22 [162]. In addition, PD-1 blockade enhanced production of IL-6, an antiviral cytokine, and inhibited B cell apoptosis [162]. 

A separate study analyzed the efficacy of those antibodies that were produced by peripheral B cells [164]. The B cells were obtained from six vaccinated patients and eight patients who spontaneously cleared a chronic HBV infection. ELISA binding analyses of B cells specific for S-HBsAg demonstrated that only 21.1% of antibodies produced by B cells from vaccinated patients demonstrated high affinity, as compared to 55.2% of antibodies from spontaneous clearers [164]. When assessing their neutralizing capability, 61% of the anti-HBs antibodies from S-HBsAg-specific B cells overall were capable of neutralizing HBV. 69% of those derived from spontaneous clearers were neutralizing [164]. Approximately half of the neutralizing antibodies were capable of cross-reacting with HBsAg from nine genotypes (A-I). Some of these neutralizing antibodies were also capable of inhibiting HDV infection [164]. The potent neutralizer Bc1.187 was tested against and neutralized HBV genotypes A-D in vitro. Furthermore, injection of a chimeric version of the Bc1.187 antibody in infected mice resulted in decreases in both HBsAg and HBV DNA. Levels remained suppressed for two weeks following antibody injection [164]. 

Salimzadeh et al. recently analyzed how HBV infection influences the composition of the B cell compartment as well as the effect of PD-1 blockade on virus-specific B cell function [165]. They used the fluorochromes DyLight 550 and DyLight 650 to label HBsAg derived from genotype A viruses. These fluorochromes were then used to bind anti-HBsAg B cell receptors. The results showed that HBsAg-specific B cells comprise on average 0.079% of all B cells in those infected with chronic HBV and 0.053% in those with acute infections. HBsAg-specific B cell levels were not associated with HBsAg levels, HBV DNA, or ALT levels [165]. Interestingly, the frequency of plasmablasts remained low during acute HBV infection in 5 of 6 patients examined. Levels of HBsAg-specific B cells were noted to be comparable to those seen in resolved HBV infection, indicating that the quantity of circulating B cells specific to HBsAg are not indicative of the degree to which the HBV infection is controlled [165]. The authors also demonstrated impaired antibody production of HBsAg-specific B cells in chronically infected patients when compared to vaccinated controls. A similar trend was noted in patients acutely infected with HBV; however, B cells collected from patients who had undergone seroconversion were able to produce antibodies against the hepatitis B surface antigen (anti-HBs). This data indicates that the ability of HBsAg-specific B cells to secrete antibodies is impacted by HBsAg [165]. 

The same set of experiments showed that chronically infected patients had significantly more AtM receptors, both in general and specific for HBsAg, than the vaccinated control sample [165]. This is in accordance with previously discussed experiments [162]. Serum HBsAg was not correlated with overall atypical memory B cell frequency, although HBV DNA was [165]. To further show that HBV infection altered the global B cell population, the authors analyzed changes in gene expression in the B cells of patients infected with HBV. Differential expression was seen in numerous genes, including those involved in B cell activation, proliferation, and differentiation [165]. Inhibiting PD-1, a gene that was shown to be upregulated in AtM B cells during chronic HBV infection, stimulated the proliferation of anti-HBs cells in infected individuals [165]. 

An interesting aspect of HBV replication is the production of several types of non-infectious subviral particles (SVPs) in great excess to infectious virions [166]. These SVPs incorporate HBsAg [166] and were recently shown to have a direct impact on the humoral immune response [167]. After fractionating HBsAg-containing serum using a Nycodenz gradient to purify viral particles and SVPs, Rydell et al. performed serial dilutions of samples with 10,000 and 30,000 IU/L of anti-HBs that were incubated with 3.2 GE/cell of HBV DNA from serum with HBsAg concentrations of 500 IU/mL (unfractionated serum), 10 IU/mL (enriched with viral particles), and a 1:1 mixture of solutions with HBsAg concentrations of 10 IU/mL and 100 IU/mL. The ability of anti-HBs to neutralize viral particles was significantly diminished in mixtures containing higher HBsAg concentrations. The most significant antibody inhibition was seen with unfractionated serum, which had the highest HBsAg concentration [167]. Figure 3 summarizes the effects of HBV on B and T cells. 

## 6. Antigenicity of HBsAg 

HBsAg contains a region known as the “a” determinant [168,169], a key immunogenic segment with multiple epitopes [170,171]. Importantly, many neutralizing antibodies target this region [169]. Mutations in this region can lead to decreased antigenicity and immune escape, even in the setting of otherwise immune patients [172]. The most prevalent immune escape mutation includes arginine replacing glycine at amino acid 145. Populations in which this mutation is seen include those with anti-HBs, including those treated with hepatitis B immune globulin (HBIg) and vaccinated children born to mothers with chronic HBV [173].

Antigenicity can also be impacted by mutations elsewhere [174]. For example, Hossain et al. isolated and characterized the viral genetic material from a Bhangladeshi patient with a genotype C HBV infection. Nine mutations in the preS1 region of L-HBsAg were identified. L-HBsAg isolated from this patient demonstrated decreased binding capacity with preS1 ELISA, and anti-preS1 antibody was unable to detect L-HBsAg in the mutated sample [174]. Additionally, mutations in several cysteine residues in S-HBsAg have been shown to impact the antigenicity of the small subunit [175]. 

As was mentioned above, HBsAg is composed of small, middle, and large HBsAg. These discrete subunits have differing levels of antigenicity. A comparison of plant-derived M-HBsAg and S-HBsAg showed that M-HBsAg stimulated greater anti-HBs production following intramuscular administration to mice [176]. IgG1 was the dominant antibody generated by both subunits. S-HBsAg was able to stimulate a stronger IgG2a response, although IgG2B responses were equivalent. Only M-HBsAg was able to stimulate a humoral response against PreS2 [176]. 

## 7. HBV and Immune Suppression

Immunosuppression has a variety of causes, such as HIV [177] and chemotherapy [178]. Immunosuppression affects the natural course of HBV infection and can lead to impaired virus-specific immune response to infection or vaccination [179,180] and viral reactivation [181]. Indeed, it is worth testing immunosuppressed patients for HBV infection status and offering vaccination if no serologic indications of prior exposure are documented [182]. 

The relationship between HBV and HIV is interesting. Approximately 10% of all people infected with HIV are also afflicted with HBV [183]. Coinfection is associated with increased risks of chronic hepatitis infection, liver fibrosis, and hepatocellular carcinoma [183]. Contrary to what one may expect, several studies have demonstrated that HBV infection is more easily managed pharmacologically and has a higher rate of seroconversion in patients being treated for a concurrent HIV infection than in those without [183,184]. For example, 29 patients of 284 analyzed (10.2%) were found to be HBsAg negative following antiretroviral therapy. 12 of those patients developed anti-HBs [184]. This is higher than the rate of seroconversion in patients with HBV monoinfection [185,186]. Functional cure was associated with a baseline CD4 count of <350 cells/mm^3^, female sex, and lower baseline HBV DNA [184]. Another study assessing coinfected individuals found that 1419 (2.4%) of 59,829 HIV-infected patients were HBsAg positive for at least 6 months [187]. 97 patients cleared HBsAg, and 67 of those developed anti-HBs. Early initiation of HBV treatment, a longer duration of HBV therapy, tenofovir disoproxil fumarate therapy, an African origin (versus Caucasian), and patients whose lowest CD4 count remained relatively high were associated with HBsAg clearance [187]. Comparisons were made between tenofovir disoproxil fumarate, lamivudine, and emtricitabine alone or in combination [187]. 

While this trend is not seen in the setting of untreated HIV/HBV coinfection [188], improved control is seen in patients with an acute HIV infection superimposed on a chronic HBV infection [189]. Song et al. recently delved into the mechanisms through which this occurs [190]. Flow cytometry demonstrated that co-infected patients have increased levels of circulating natural killer (NK) cells [190]. NK cells are capable of expressing the inhibitory receptor NKG2A [191] and the activating receptor NKG2C [192,193]. While NKG2A was found to be more prominent in patients with HBV infection when compared to coinfected individuals, HIV-infected individuals, and healthy controls, the frequency of NKG2C was upregulated in patients with HIV/HBV coinfection and HIV-infected individuals compared to those infected with HBV alone [190]. There was also an increase in the proportion of CD56^neg^ NKG2C^+^, CD56^dim^ NKG2C^+^, and CD56^bri^ NKG2C^+^ NK cells as well as a decrease in CD56^dim^ NKG2A^+^ NK cells in coinfected individuals when compared to HBV-infected individuals. Analyzing co-expression of these molecules showed increased NKG2A^+^NKG2C^−^ and decreased NKG2A^−^NKG2C^+^ NK cells in HBV-infected individuals when compared to co-infected individuals [190]. Importantly, degranulation assays demonstrated increased expression of CD107a in coinfected individuals. There was increased cytotoxic activity and IFN-γ production in coinfected individuals relative to HBV monoinfection, and IL-10 production was decreased [190]. Analysis of viral load showed a positive correlation between NKG2A^+^ NK cells and NKG2A^+^NKG2C^−^ NK and viral load. In addition, there was a negative correlation between NKG2C^+^ NK cells and NKG2A^−^NKG2C^+^ NK as well as IFN-γ^+^ NK cells and viral load [190]. 

Cancer, and specifically cancer therapy, can result in immunosuppression and subsequent HBV reactivation [194]. For example, Kusumoto et al. [195] analyzed 2797 patients enrolled in the GALLIUM [196] and GOYA [197] trials, which are clinical trials assessing the efficacy of obinutuzumab compared to rituximab in previously untreated lymphoma for risks of HBV reactivation and HBV-associated hepatitis. 326 patients had a resolved HBV infection, of whom 94 received anti-HBV nucleos(t)ide therapy (NA) prophylactically. 25 of the 232 patients (10.8%) who did not receive prophylactic NA experienced HBV reactivation, of whom nine did not experience reactivation until after the cessation of chemotherapy. 2 of the 94 patients (2.1%) who received prophylactic NA had HBV reactivation. None of the patients with HBV reactivation developed hepatitis [195]. Risks for reactivation in the setting of immunochemotherapy include older age, type of lymphoma (patients with diffuse B cell lymphoma were most likely to experience reactivation), absence of anti-HBs, and detectable HBV DNA prior to initiating therapy. Prophylactic NAT was associated with a decreased risk of reactivation [195]. 

A similar study assessed the risk of HBV reactivation in the setting of PD-1 inhibition, PD-L1 inhibition, or combination therapy [198]. 114 patients with a variety of malignancies, including nasopharyngeal carcinoma, HCC, melanoma, and non-small cell lung cancer, were analyzed. 35 had detectable HBV DNA prior to receiving treatment. Notably, 85 patients received antiviral prophylaxis, including 30 of the patients with detectable HBV DNA at baseline. Six patients of the 114 (5.3%) developed HBV reactivation, one of which began after discontinuing immunotherapy. Five of the patients went on to develop HBV-related hepatitis. Five of the patients with HBV reactivation did not receive antiviral prophylaxis, while the sixth did [198]. Interestingly, baseline HBV DNA was not associated with reactivation risk in this study [198]. In agreement with the previously discussed trial [195], antiviral prophylaxis was associated with a significantly lower risk of HBV reactivation. 

## 8. S-HDAg, L-HDAg, and the Immune System

As was discussed above, HDV encodes no protein besides the two isoforms of HDAg, L-HDAg and S-HDAg. This presents a relatively limited number of antigens for the immune system to recognize [199]. Indeed, patient CD4+ and CD8+ T cells were shown to recognize an average of only 1.2 out of 21 overlapping 20-mer HDV-specific peptides after incubation with the overlapping peptides and IL-2 [200]. The peptides examined included overlapping peptides that cumulatively include the entirety of L-HDAg. Incubation lasted for 10–12 days. On average, CD4+ T cells accounted for 0.75 responses while CD8+ T cells accounted for 0.44 responses [200]. 

HLA molecules are responsible for presenting peptides to T cells. The allelic variant has implications regarding which amino acid sequences can be presented [201]. Assessment of three HLA-B*27 negative patients and three HLA-B*27 positive patients showed that T cell responses to L-HDAg were only noted in the HLA-B*27 positive patients [202]. Subsequent analysis revealed the presence of a key antigenic region in L-HDAg including amino acids 98–113. Two peptide ligands (amino acids 99–108 and 103–112) of HLA-B*27were noted in this segment. Notably, chronically infected patients not expressing HLA-B27 were unable to mount an immune response to that section of HDAg [202]. However, it is worth noting that HDV may not provide a particularly strong stimulus to T cells given the relatively weak T cell responses noted in previous experiments [200,203]. 

A pressing issue that contributes to HDV chronicity is the presence of mutations that allow HDV to evade the immune system [202]. 104 patients with HBV/HDV coinfection were analyzed to determine whether viral polymorphisms in L-HDAg led to immune escape in the presence of specific HLA class I molecules. 21 were found. 16 of the 21 were associated with HLA-B while the remaining 5 were associated with HLA-A. Of note, significant polymorphisms were only seen in the central region of HDAg, and none were observed in the L-HDAg-specific C terminal extension [204]. Some CD8+ T cells specific for prototype epitopes could respond to variant epitopes, but CD8+ T cells specific for the variant epitopes were unable to respond well to either prototype or variant epitopes, indicating successful immune evasion [204]. A separate set of experiments demonstrated that a mutation in L-HDAg, K106M, can lead to viral escape by altering the L-HDAg epitope comprised of the amino acids in positions 103–112 [202]. These escape variants hinder functional memory cells from successfully eliminating HDV infection [205]. 

Notably, both forms of HDAg are poor B cell stimulators, with limited to no antibody response generated following injection of plasmids expressing HDAg and L-HDAg, even following co-injection with plasmids expressing GM-CSF, IL-12, and IL-18 [206]. While injecting a woodchuck model with HDAg B cell epitopes did generate an immune response with HDAg-specific antibodies, it did not confer protection from HDV infection [207,208]. Table 2 summarizes the interactions between the immune system and HBV/HDV. 

## 9. Therapies for HBV and HDV

In 2016, the World Health Organization outlined their goal of eliminating viral hepatitis with a focus on HBV and hepatitis C virus (HCV) by the year 2030 [209]. According to the Chronic Hepatitis B Virus Infection: Developing Drugs for Treatment Guidance for Industry published by the U.S. Department of Health and Human Services Food and Drug Administration Center for Drug Evaluation and Research (CDER), the present therapies yield suppression of HBV DNA during treatment, but rates of HBsAg loss with or without seroconversion to anti-HBsAg stay at a decreased level [210]. One of the main tools available in this endeavor is a functional vaccine for HBV. However, low vaccination rates in some countries have facilitated the persistence of HBV in the general population [211]. In the United States, infants are typically given the first of three doses at birth, the second dose between one to two months of age, and the third and final dose between 6–18 months of age [212]. Additionally, the vaccine is available for children, adolescents, and adults who were not previously vaccinated during infancy [213]. The vaccine carries recombinant HBsAg synthesized in yeast [214,215]. The antibodies produced by the vaccine target HBsAg and lead to broad immunity against all genotypes of the virus [216]. In the event of accidental exposure to HBV, post exposure prophylaxis (PEP) with the hepatitis B vaccine administered as soon as possible can prevent HBV infection and subsequent development of chronic infection or liver disease [217,218]. In certain cases, the PEP can be combined with hepatitis B immune globulin to confer added protection [217,218]. Vaccination is typically only recommended for HBV seronegative patients [219,220]. There is no role for vaccination in already infected patients per the Centers for Disease Control [220]. Currently, there is no vaccine to protect against HDV. However, preventing HBV infection through the use of the HBV vaccine can protect against future HDV infections [221]. 

The current recommended antiviral treatment protocol for chronic HBV infection, regardless of severity of liver disease, is the long-term administration of a nucleos(t)ide analogs (NAs) with a high barrier to resistance. Such NAs include entecavir, tenofovir-disoproxil-fumarate, and tenofovir alafenamide [219]. NAs control HBV replication by suppressing HBV reverse transcriptase but may not reduce HBsAg or restore the immune response [222]. Interestingly, they have been linked to a decreased risk of developing HCC in both cirrhotic and non-cirrhotic patients [223,224]. Discontinuation of NAs often leads to relapse since it does not target cccDNA [225] or integrated HBV DNA. Because HBsAg loss is associated with further decrease in HCC risk compared to HBV DNA suppression without HBsAg loss, the former (termed functional cure) is considered a desirable clinical endpoint [226]. It should be noted that there is a difference between achieving a functional cure and a complete cure (or eradication) with regards to HBV infection. Clinical guidelines define a functional cure as HBsAg loss (i.e., undetectable HBsAg with current assays) with or without anti-HBs and undetectable HBV DNA 6 months after completing treatment [227]. This type of cure is rarely achieved with the use of NAs and/or pegylated interferon alpha-2a (PEG-IFNα) therapy but is more likely in combination with other investigational drugs such as nucleic acid polymers and NAPs [228]. Future combination regimens for HBV cure are ongoing, with the goal to combine multiple agents that inhibit HBV replication (e.g., NAs and capsid inhibitors), translation inhibitors that suppress HBsAg (e.g., silencing RNA’s and antisense oligonucleotide) and restore immune control (e.g., NAPs, PEG-IFNα, and PDL1 inhibitor). For a comprehensive list of anti-HBV drugs in trials, see https://www.hepb.org/treatment-and-management/drug-watch/ (accessed on 11 November 2022). 

Treatment for HDV is very promising in spite of the lack of virus-specific enzymes, which prevent it from being directly targeted to inhibit its replication [229]. The primary goal of treatment is the suppression of HDV replication (or even eradication) and/or accompanying improvements in ALT and reduction in hepatic necroinflammation [80]. There is no FDA-approved antiviral therapy for HDV. The recommended antiviral treatment against HDV is PEG-IFNα [230,231]. While there is little data regarding its mechanism of action, PEG-IFNα has been predicted to block HDV production in patients [99]. This is in agreement with the decrease in genomic and anti-genomic HDV RNA levels reported in humanized mice [232]. Unfortunately, PEG-IFNα may cause severe side effects and is contraindicated in patients with advanced cases of liver cirrhosis [230]. Additionally, PEG-IFNα treatment rarely results in HBsAg loss [233]. Anti-HDV drugs such as lonafarnib [100,234], pegylated interferon lambda [235], the entry HDV/HBV inhibitor bulevertide [236,237,238] that reduce HDV levels are in development [239]. Thus far, the only anti-HBV treatment that affects HBsAg production is NAPs, which is accompanied by declines in both HBsAg and HDV RNA [98,240]. For a comprehensive list of anti-HDV drugs in trials, see https://www.hepb.org/treatment-and-management/drug-watch/ (accessed on 11 November 2022).

## 10. Concluding Remarks

HBV and HDV are highly prevalent viruses, with HBV estimated to infect almost 300 million people [241], of whom HDV is estimated to infect 12–72 million people worldwide. They can cause significant hepatic disease, including cirrhosis, fulminant hepatitis, and hepatocellular carcinoma, and represent a significant cause of global mortality. The host immune response is thought to play a significant role in the pathogenesis of both HBV and HDV. 

The viruses and antigens produced by these viruses impair T and B cells. Particularly prominent are changes in T and B cell phenotypes as demonstrated by alterations in the frequency and function of several subpopulations of T and B cells, such as changes in the circulating levels of Tfh cells and AtM cells. Particularly prominent are changes leading to T and B cell inhibition, including expression of inhibitory receptors such as PD-1 and FcRL5. The mainstay of treatment for these viruses is the HBV vaccine, which provides immunity against HBV infection (and HDV by extension since HDV is a satellite virus of HBV). For those who have contracted HBV or HBV/HDV, there are new and promising therapies in addition to existing therapies such as pegylated IFN-α and NAs. Ongoing research will continue to elucidate novel therapies and therapeutic targets. For example, the discovery of a key viral fusion peptide in L-HBsAg indicates that viral fusion inhibitors may represent a currently unexplored avenue of treatment. This is particularly interesting given that fusion inhibitors such as ginkgolic acid have shown promise in multiple types of viruses, including herpes simplex virus, cytomegalovirus, zika virus, and coronavirus [242,243,244].

In this article, we provided an overview of the general characteristics of HBV and HDV infections. We also discussed interactions between HBV/HDV and the immune system, with a focus on how HBsAg and HDAg impact immune responses. We concluded with a brief overview of recent advances in HBV/HDV therapeutics. 

These viruses continue to represent a major challenge to public health. While there continue to be difficulties in achieving the World Health Organization’s goal of eliminating viral hepatitis by 2030, significant progress is being made [245,246,247,248]. Until such time as the World Health Organization’s goals have been achieved, a solid understanding of the interaction between HBV, HDV, and the immune system remains essential. 

## Figures and Tables

**Figure 1 ijms-23-15973-f001:**
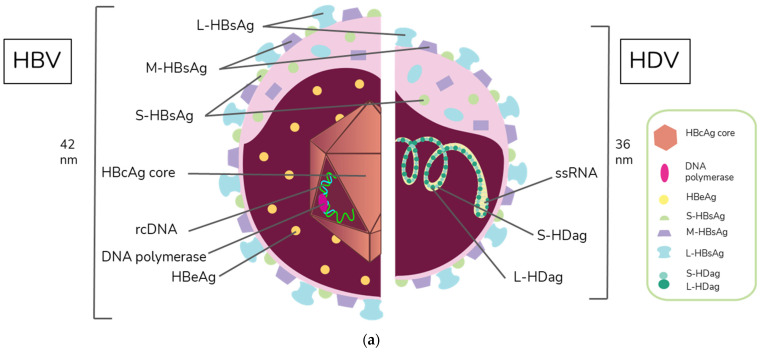
Virion and genome structures of hepatitis B virus (HBV) and hepatitis D virus (HDV). (**a**) Virion Structures of Hepatitis B Virus (HBV) and Hepatitis Delta Virus (HDV). Both HBV and HDV are enveloped in HBV surface proteins designated as small (S-HBsAg), middle (M-HBsAg), and large (L-HBsAg). The HBV relaxed-circular DNA (rcDNA) is partially double-stranded, with a full minus strand and incomplete plus strand, and a surrounding icosahedral capsid comprised of the core antigen. HBeAg proteins are present between the envelope and capsid. The HDV virion contains a single-stranded RNA genome that is complexed with small and large delta antigens (S-HDAg and L-HDAg, respectively). (**b**) Genomic Structure of HBV and HDV. HBV is a partially dsDNA virus composed of four distinct open reading frames (ORFs): polymerase, S, core, and X. There is significant overlap between the different ORFs. The RNA of both the single-stranded RNA genome and the antigenome contain self-cleaving ribozymes [53]. Unedited antigenome ultimately codes for S-HDAg. The cellular enzyme adenosine deaminase acting on RNA 1 (ADAR1) ADAR1 can modify the antigenome to generate the genetic code for L-HDAg [54].

**Figure 2 ijms-23-15973-f002:**
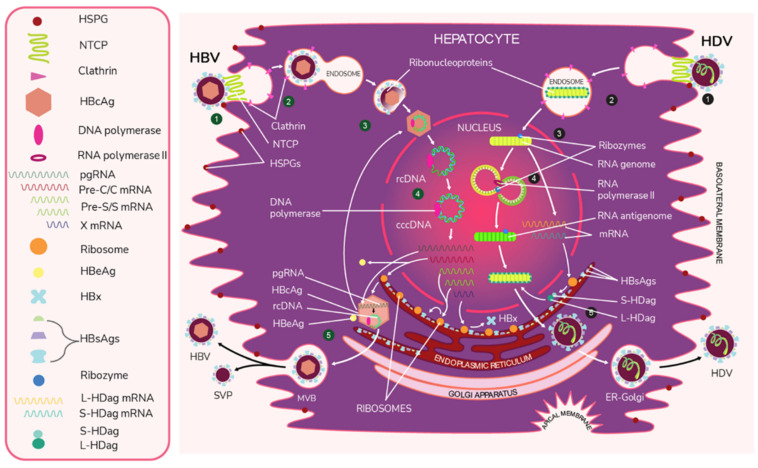
The viral life cycles of HBV and HDV. HBV (green numbers): (1) HBV attaches to the hepatocyte by first binding to host heparan sulfate proteoglycans (HSPGs) and then binding to the liver cell entry receptor sodium taurocholate cotransporting polypeptide (NTCP) with the pre-S1 domain of L-HBsAg. (2) HBV enters the cell via the clathrin-mediated endocytosis pathway through clathrin protein interactions with L-HbsAg on the HBV membrane. (3) The viral envelope fuses to the early endosome, and the core protein of the exposed nucleocapsid binds to the host nucleus and releases its viral DNA and DNA polymerase. * (4) The virus begins replication by first completing its partially double-stranded relaxed circular (rcDNA) using DNA polymerase to create full dsDNA in the form of covalently closed circular DNA (cccDNA). The DNA is then transcribed by host RNA polymerase II into pregenomic RNA (pgRNA) and 4 mRNAs. The 4 mRNAs are translated by host ribozymes. The “Pre-C/C” mRNA is translated into HBcAg core and HBeAg, the “Pre-S” and “S” mRNAs into HBV surface proteins, and “X” mRNA into HBx protein, which is involved in infection and replication. The pgRNA is transported to the cytoplasm, where it is encapsidated by the HBcAg core. Inside this preliminary nucleocapsid, HBV polymerase reverse transcribes pgRNA into negative-sense DNA, which undergoes synthesis into partially double-stranded rcDNA. The completed nucleocapsid either travels back to the nucleus to create additional virions or buds from the ER membrane, acquiring an envelope that is embedded with HBsAg. (5) Along with the infectious virions, empty subviral particles (SVPs) also bud out from the ER. The exiting viral and subviral particles pass through the Golgi apparatus into multivesicular bodies (MVBs), which then bind to the hepatocyte envelope and are released via exocytosis. HDV (black numbers): (1) HDV attaches to the host cell through low-specificity binding to HSPGs and high-specificity binding to NTCP with the viral L-HBsAg pre-S1 domain. (2) The virus enters the cell. (3) The viral envelope is uncoated and the ribonucleoprotein targets the hepatocyte nucleus. * (4) In a rolling circle manner, the viral RNA genome is transcribed into the antigenome, and the antigenome serves as a template for new genome transcripts, both of which self-cleave and ligate to reform the circular RNA. The genome also produces S-HDAg and L-HDAg mRNA from the same open reading frame. This process produces approximately 300,000 genome and a smaller number of antigenome molecules. (5) The translated protein products re-enter the nucleus to complete replication and particle assembly. Fully assembled HDV virions pass through the ER-Golgi complex and exit the host cell. * The exact uncoating mechanisms of the envelope and nucleocapsid are not fully understood.

**Figure 3 ijms-23-15973-f003:**
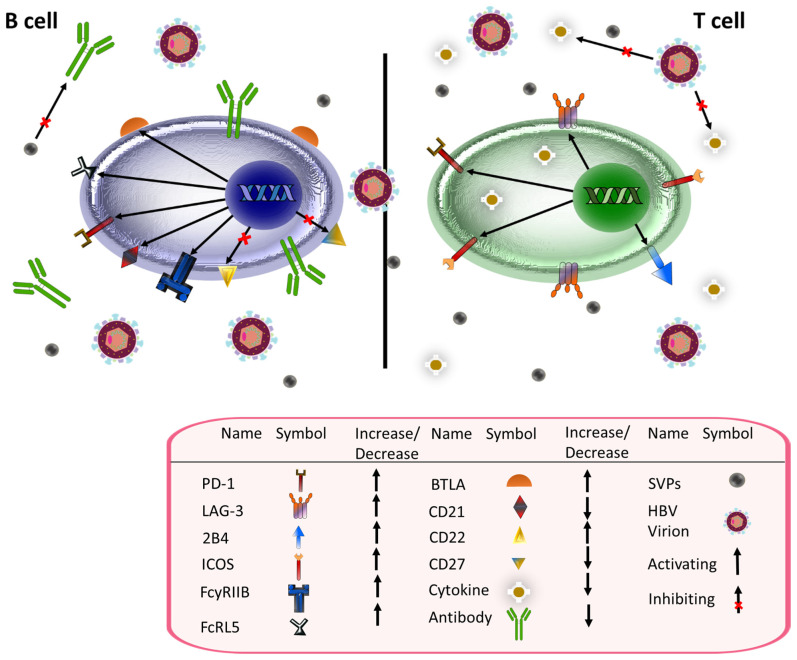
Chronic HBV-induced phenotypic changes in B (**left**) and T (**right**) cells. HBV infection results in several phenotypic changes in B and T cells. In B cells, it inhibits expression of CD21 and CD27 and increases the proportion of atypical memory B cells, which are dysfunctional B cells that express higher levels of inhibitory receptors such as PD-1 and FcRL5. In addition, HBV infection inhibits anti-HBs antibody production. Subviral particles (SVPs), the majority of which include HBsAg, adversely impact the efficacy of circulating antibodies. Like with B cells, HBV stimulates T cell expression of inhibitory receptors including PD-1 and LAG-3. In addition, regulatory T cell (Treg) markers such as ICOS are elevated. Cytokine and interleukin expression is also impaired. In the above figure, black arrows represent upregulation while black arrows with an x represent inhibition.

**Table 1 ijms-23-15973-t001:** HDV RNA and HBV DNA profiles during chronic infection.

Study|Viral Profile	HDV Dominant	HBV Dominant	HDV/HBV Equivalent *	HBeAg (+/−)	HDV RNA Assay **	Reference
Individual Patient Data						
Yurdaydin et al., 2018	17/20	0/20	3/20	3 (+) 12 (−) 5 (N/A)	In-house PCR. (LLoQ) 70 IU/mL and (LLoD) 50 IU/mL	[97]
Shekhtman et al., 2020	12/12	0/12	0/12	12 (−)	Robogene MK I LLoQ 3.26 log U/mL	[98]
Schaper et al., 2010	13/25	6/25	6/25	5 (+) 20 (−)	rt-RT-PCR in the LightCyclerTM system	[94]
Guedj et al., 2014	11/12	0/12	1/12	9 (−) 3 (+)	real-time PCR (external) (LLOQ): 100 (GE/mL)	[99]
Koh et al., 2015	8/9	0/9	1/9	9 (−)	qRT-PCR LLoD 70 IU/mL	[100]
Patient Data (Cohort Mean)						
Castelnau et al., 2006	14	NA	NA	All negative	RT-PCR 100 copies/mL	[101]
Manesis et al., 2007	53	NA	NA	All negative	PCR TaqMan Universal 100 copies/mL of serum and the linearity of quantification ranged from 103 to 109 copies/mL.	[102]
Zachou et al., 2009 ***	73	NA	NA	(+) 12 (−) 68	RT-PCR	[103]
Wedemeyer et al., 2011	90	NA	NA	14 (+) 76 (−)	TaqMan One Step PCR Master Mix LLOD of HDV RNA was 120 copies/mL of serum	[104]
Braga et al., 2014 (64 patients–only % given)	56.3%	3.1%	40.6%	All non-reactive	qRT-PCR Sensitivity not listed.	[95]
Koh et al., 2015	14	NA	NA	All negative	qRT-PCR LLoD 70 IU/mL	[100]
Bogomolov et al., 2016	8	0	16	All Negative	Amplisense HDV-FL During treatment: qPCR: limit of detection (LOD) 15 copies/ml	[105]
Yurdaydin et al., 2007	39	0	0	(+) 2 (−) 37	RT-PCR Sensitivity: 100,000 copies/mL	[106]

* In Braga et al. [95], HDV/HBV equivalence is defined as less than 2 Log10 difference in viral load. For consistency, two of the HBV dominant patients in Schaper et al. were adjusted to HBV/HDV equivalent profiles based on a criterion of Log 2 or less difference in HBV DNA and HDV RNA. ** Differences in assay sensitivities are a limitation of the data analyzed. Analysis and description are limited to each journal article data set, not across all data sets. *** 7 patients dropped out of the study.

**Table 2 ijms-23-15973-t002:** Summary of Interactions between the immune system and HBV/HDV.

Virus/Viral Component	Effect on Immune System	Reference
HBV	Increased inhibitory CD8+ receptors including PD-1, LAG-3 as well as inhibition of IFN-γ, IL-2 and TNF-α secretion	[127,140]
HBV	Increased frequency of central memory CD4+ T cells	[140]
HBV	Inhibition of germinal center formation and Tfh expansion	[146]
HBV	Inhibit Tfh secretion of IL-21	[153]
HBV	Increased plasmablasts, plasma cells, regulatory B cells, and CD19+ B cells	[153]
HBV	Inhibition of HBsAg-specific B cell maturation	[161,165]
HBV	Decreased expression of genes involved in class switching, increased expression of genes in endothelial adhesion, homeostasis, apoptosis	[161]
HBV	Decrease antibody production	[161,162]
HBV	Immune Escape	[172,173]
HBV	Increased atypical memory B cells	[162,165]
High HBsAg	Elevated PD-1 on CD4+ T cells, and increased levels of CD4+ T cells with high PD-1 levels	[140]
High HBsAg	Fewer T cells secreting multiple types of cytokines	[140]
High HBsAg	higher PD-1 expression on EMRA T cells	[140]
High HBsAg	Increased expression of Inhibitory receptors (FcRL5, coexpression of 2B4 and PD-1) on CD4+ T cells	[140]
High HBsAg	Inhibit antibody efficacy	[167]
HBsAg	Increased number of T cells specific for viral envelope	[120]
HDV	Lack of B/T cell stimulation	[199,200,203,206,207,208]
HDV	Escape mutations	[202,204,205]
HDV	HLA and immune response	[202,204]

## Data Availability

The sources referenced in this work are available from PubMed, Google Scholar, Fields Virology 6th edition, the World Health Organization, the Center for Disease Control and Prevention, and the U.S. Food and Drug Administration.

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
