# Peer review of "Hepatitis B and Hepatitis D Viruses: A Comprehensive Update with an Immunological Focus"

_ijms, 2022, doi:10.3390/ijms232415973_

Round 1

Reviewer 1 Report

The review by Sausen et al. provides a comprehensive overview of HBV and HDV infection, including pathological features, immunology and infection in the context of immune deficiency and cancer. The work provides a comprehensive update of HBV and HDV virology, as well as current treatment protocols and viral pathogenesis. I have some minor suggestions for improvement prior to acceptance:

1. The review provides such a comprehensive update, I suggest modifying the title to encompass all aspects of the subject covered, which goes beyond the interactions of HBV and HDV with the immune system.

2.Lines 320 and 333: Change ‘spotty’ necrosis to ‘multifocal’ or ‘random multifocal’, or another recognised pathological term.

Author Response

We thank the reviewer for his kind comments regarding our paper, and we appreciate the opportunity to improve the manuscript.

  1. The review provides such a comprehensive update, I suggest modifying the title to encompass all aspects of the subject covered, which goes beyond the interactions of HBV and HDV with the immune system.
  • The title was changed to ‘Hepatitis B and Hepatitis D Viruses: A Comprehensive Update with an Immunological Focus’. In addition, the abstract and key words were updated to reflect the breadth of the article as well.

        2. Lines 320 and 333: Change ‘spotty’ necrosis to ‘multifocal’ or ‘random                  multifocal’, or another recognized pathological term.

  • All instances of spotty necrosis were changed to ‘focal necrosis’.

Reviewer 2 Report

This is one of the best comprehensive review on both HBV and HDV.This review will be highly educational for all hepatologists and virologists supplied with updated data on both viruses.

 The authors have conducted a commendable review on these two viruses and their concurrent infection. The severity of these two viruses whether coinfected or superinfected, was individually described and discussed.  The importance of the host immune responses was further elaborated, in particular, to the individual antigenic HBsAgs (S-HBsAg, M-HBsAg, L-HBsAg). Furthermore, HDV antigens, L-HDAg and S-HDAg  and the host immune responses were reviewed, and the therapeutic approach was presented.  These data are of high significance and will be considered in designing more effective treatment for these viruses.  The references were excellent. The tables and figures are also very good. 

I trust this review will be highly educational for the hepatologists and virologists. 

The authors are highly commended for their work.

Author Response

We thank the reviewer for his kind comments regarding our paper.